# 🌱Gecko: A Simulation Environment to Ground Agent Tool Calls with Stateful Feedback for Refinement

## Abstract

The ability to use tools is fundamental to large language model (LLM) agents. However, when solving complex tasks, current LLMs are prone to incorrect tool selection and invalid tool-call arguments. Although letting LLMs iteratively refine the tool-call sequence using execution results from real tools can help, repeated testing on real tools can be expensive and lead to unintended side effects. To improve LLM tool calls while addressing the issues caused by using real tools for refinement, we introduce Gecko[1], an environment that simulates tool responses using a combination of rules and LLMs. Specifically, Gecko checks the validity of tool calls including input arguments and tool names, synthesizes reasonable responses that adhere to the output schema, and assesses whether all task objectives have been achieved. Such feedback provided by Gecko allows LLMs to refine their tool calls, forming a simple yet effective test-time scaling method named GATS. In addition, we design an automated API schema converter so that Gecko can quickly integrate and simulate a large number of tools. On BFCL and $\tau^2$-bench, our test-time scaling method GATS enabled by Gecko consistently improves tool calling performance of existing LLMs including GPT-4o and GPT-5 (Fig. 1) and yields new state of the art. We further discuss working mechanisms of our method and share rosy future possibilities.

## 1 Introduction

Building agent systems using LLMs to solve complex tasks has become increasingly popular. In this mission, it is critical to let LLMs be able to use external tools, such as `get_weather`

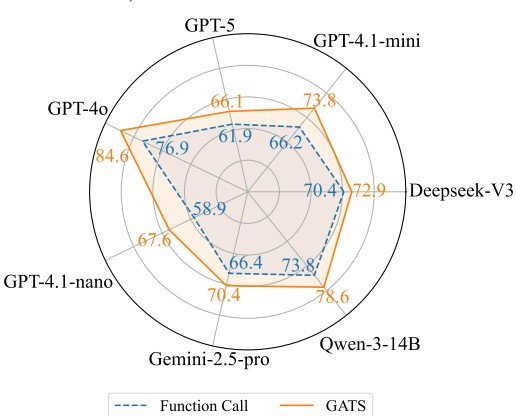

Figure 1: GATS consistently improves the performance of different LLMs on BFCL.

and `fetch_stock_data`. While there exist strong LLMs such as GPT-4o OpenAI et al. (2024), Qwen3 Yang et al. (2025), and xLAM-2 Prabhakar et al. (2025), because of long contexts, high task complexity, and rigid tool definitions, it is still challenging for these LLMs to select suitable tools and give accurate arguments Kate et al. (2025); Huang et al. (2024).

To improve the tool-calling capabilities of LLMs, some existing methods let LLMs use real tools and use the execution results as feedback to refine LLM instructions Singh et al. (2025); Shi et al. (2024); Kang et al. (2025); Li et al. (2025). While this strategy improves tool call accuracy, it is limited because using real tools can be costly (*e.g.*, RapidAPI charges a service fee) and may yield undesirable consequences Li & Fung (2025). For example, inappropriate execution of `Tweet_Post` during inference may leak information irrelevant to the task, even if the post is deleted afterward.

---

[1]Gecko comes from keywords a**ge**nt + feedba**ck** + envir**o**nment.

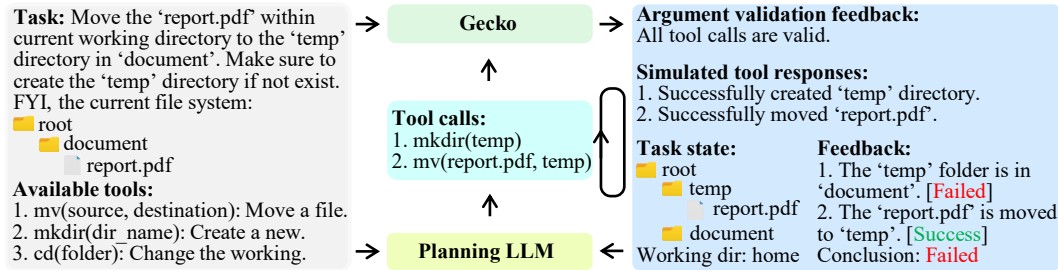

Figure 2: Overview of tool call refinement based on feedback from Gecko. The planning LLM generates tool calls based on a task and available tools. Gecko processes these calls and provides feedback on argument validation, simulated tool responses, and task state. The three types of feedback are used by the planning LLM for iterative refinement.

In this work, we aim to improve the performance of LLM[2] tool call and avoid using real tools during test-time refinement. To this end, we introduce Gecko, a comprehensive environment hosting a large number of simulated tools that produce semantically reasonable outputs and share the same input and output formats with real tools. As shown in Fig. 2, after receiving a task from user and tool calls from a planning LLM, Gecko will simulate the execution of the tool calls and provide three types of feedback. **First**, Gecko checks whether the input follows predefined formats, *e.g.*, 'input 2 is invalid because only float numbers are allowed.' This is implemented by a combination of rules and a helper LLM. **Second**, Gecko simulates tool responses. To ensure the simulated responses are semantically suitable and consistent with prior tool calls, we carefully prompt a helper LLM with the validated tool call, the tool schema, and the current task state. **Third**, Gecko uses a helper LLM to estimate the key states that reflect task progress based on the simulated tool responses. A judge LLM then assesses the inferred task state, determines whether task objectives are met, and provides task feedback on completion status and any outstanding issues.

Further, the above three types of feedback allow us to naturally design a tool-call refinement method named GATS (grounding agent test-time scaling). In this method, feedback from Gecko is sent to the planning LLM to refine the tool calls, which are then fed to Gecko to collect further feedback. This process iterates until task feedback indicates success or until it exceeds the maximum retry times. Through GATS, we observe more correct arguments and a more reasonable selection of tools.

Therefore, by grounding agent tools calls in the Gecko virtual environment, we collect useful feedback for tool-call refinement while avoiding the cost incurred by real tools.

We perform extensive evaluation on the BFCL benchmark Patil et al. (2025) and $\tau^2$-bench Barres et al. (2025), where Gecko has automatically synthesized 8,578 and 25 tools, respectively. We show that tool calls generated by many LLMs, such as GPT-4o and GPT-5, can be effectively hosted and executed in Gecko. As shown in Fig. 1 and Table 3, improvements are consistent across different agentic LLMs and across both single-turn and multi-turn tasks. For example, the overall performance of GPT-4o is improved from 76.93% to 84.62% on BFCL. We further discuss new tasks and possibilities that can be enabled through the proposed environment. In summary, this paper discusses the following main points.

- We introduce Gecko, a simulation environment which allows virtual tool use and gives informative feedback. Gecko successfully grounds tool calls generated by existing LLMs.

- By providing tool use feedback to the planning LLM, Gecko naturally allows for GATS, a test-time scaling that refines the tool calls during inference.

- We show our method brings consistent improvement to existing LLMs on BFCL and $\tau^2$-bench.

- We point out exciting insights and future directions made possible by Gecko.

---

[2]This paper use 'planning LLM' or 'agentic LLM' to describe such LLMs.

## 2 RELATED WORK

**Improving LLMs of their intrinsic tool-calling abilities.** ToolAlpaca Tang et al. (2023) fine-tunes LLMs on tool-use data generated by strong teacher models like GPT-4. ToolLLM Qin et al. (2023) collects a large number of real-world APIs and uses an automatic pipeline to construct instruction-tuning data for tool-use fine-tuning. APIGen Liu et al. (2024b) and ToolACE Liu et al. (2024a) improve the quality of synthetic tool-use data by format checking and semantic verification to improve fine-tuning. These methods merely focus on training data synthesis, while Gecko, due to its ability in simulating and grounding the tools, offers much higher flexibility. Gecko naturally supports test-time scaling while previous methods do not. Gecko also has very good potential in training data synthesis and reinforcement learning (future work, refer to Section 5).

**Test-time scaling for agentic tool use.** Existing methods use feedback loops or self-reflection grounded in *real* tool execution Shi et al. (2024); Du et al. (2024); Qiao et al. (2024); Singh et al. (2025); Li et al. (2025); Chen et al. (2025b); Shi et al. (2025); Shinn et al. (2023); Zhou et al. (2025). For example, ConAgent Shi et al. (2024) iteratively refines tool calls using feedback generated by an observation LLM from real tool failure messages. TRICE Qiao et al. (2024) combines behavior cloning with reinforcement learning guided by real tool execution feedback, teaching the model to refine its tool calls during inference. These methods rely on repeatedly calling real tools, leading to tool-call costs and potential side effects. In contrast, Gecko removes the need for real tool executions in test-time scaling. Moreover, while these methods provide feedback on correcting individual failed tool calls, without maintaining a task state, they are unable to provide task-level feedback.

**Simulation environments for agentic tool use.** Existing methods either provide fixed, domain-specific mock tools Styles et al. (2024); Liu et al. (2023); Chen et al. (2025a) or simply wrap real APIs Qin et al. (2023), which has limited general-purpose tool simulation. For example, ToolSandbox Lu et al. (2025) and BFCL Patil et al. (2025) provide a set of stateful tools whose outputs depend on history tool executions to simulate multi-turn tasks. $\tau$-bench Yao et al. (2024) and $\tau^2$-bench Barres et al. (2025) emulate conversations between a user and an agent in airline and retail scenarios. While these methods provide precise simulation on the designed use cases, they are limited by human-written tools and datasets and are hard to generalize. Therefore, they could only be used for agent evaluation rather than to improve LLM performance at test time.

## 3 THE GECKO SIMULATION ENVIRONMENT

Gecko has five components: (1) an argument validator that checks the syntactic and semantic validity of tool calls (Section 3.1); (2) a response generator that synthesizes realistic outputs for validated tool calls (Section 3.2); (3) a task state estimator that keeps track of the evolving task state (Section 3.3); (4) a task feedback generator that judges task completion and identifies remaining objectives (Section 3.4); and (5) an API schema converter that transforms new tools into OpenAPI 3.1 schemas for integration (Section 3.5).

### 3.1 ARGUMENT VALIDATOR

**Checking argument syntactic validity by manually defined rules**. Our rules verify the presence of all the required parameters and reject unsupported ones based on the argument definitions. Moreover, our rules check the input data types, *e.g.*, integer, string, or boolean. Besides, we ensure that input parameters are within the predefined range. We also have some other rules, listed in the Appendix A.3. Violations of these rules result in *error feedback*. See examples in Fig. 3(a).

**Checking argument semantic correctness by a helper LLM**. 'Semantic' means descriptions and common-sense knowledge about arguments. For example, the helper LLM rejects 'Seattle' if the input requires 'country'; the helper LLM identifies date formats (*e.g.*, yyyy/mm/dd) from the context and rejects incorrect date formats (*e.g.*, mm/dd/yy) generated by the planning LLM; it also rejects unreasonable input values implied by context, *e.g.*, a negative value in the 'age' field. When such semantic inconsistencies occur, *error feedback* will be generated. See examples in Fig. 3(b).

Table 1 presents the accuracy of argument validation on BFCL-Live-Simple. We use three metrics: true positive detection rate (correct arguments detected as correct), syntactic error detection rate, and semantic error detection rate. All the three metrics are computed based on rules. Details are

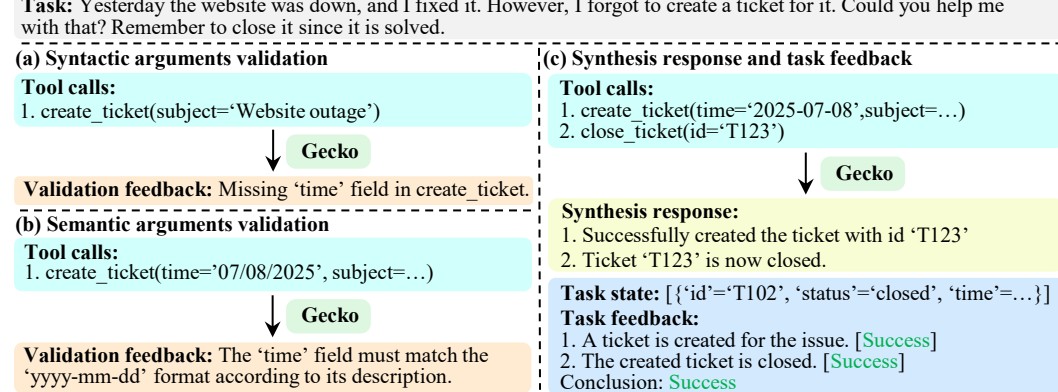

**Task:** Yesterday the website was down, and I fixed it. However, I forgot to create a ticket for it. Could you help me with that? Remember to close it since it is solved.

**(a) Syntactic arguments validation**

**Tool calls:**
1. create_ticket(subject='Website outage')

→ **Gecko**

**Validation feedback:** Missing 'time' field in create_ticket.

**(b) Semantic arguments validation**

**Tool calls:**
1. create_ticket(time='07/08/2025', subject=…)

→ **Gecko**

**Validation feedback:** The 'time' field must match the 'yyyy-mm-dd' format according to its description.

**(c) Synthesis response and task feedback**

**Tool calls:**
1. create_ticket(time='2025-07-08',subject=…)
2. close_ticket(id='T123')

→ **Gecko**

**Synthesis response:**
1. Successfully created the ticket with id 'T123'
2. Ticket 'T123' is now closed.

**Task state:** [{'id'='T102', 'status'='closed', 'time'=…}]
**Task feedback:**
1. A ticket is created for the issue. [Success]
2. The created ticket is closed. [Success]
Conclusion: Success

Figure 3: Examples of feedback provided by Gecko through (a) syntactic argument validation, (b) semantic argument validation, and (c) synthetic responses, task state, and task feedback. The validation feedback (orange), synthetic response (yellow), and task state and feedback (blue) will then be fed to the planning LLM during test-time scaling.

| Detection rate | GPT-4o | GPT-4.1 nano | Qwen2.5 7B |
|---|---|---|---|
| True positive | 100% | 100% | 100% |
| Syntactic errors | 100% | 100% | 100% |
| Semantic errors | 71% | 69% | 63% |

Table 1: Accuracy of argument validation on BFCL-Live-Simple. 'True positive' is the rate where correct arguments are determined as correct. We also show percentage of syntactic errors and semantic errors being detected. As helper LLMs, GPT-4o, GPT-4.1-nano and Qwen-2.5-7B are evaluated.

| | True positive | True negative |
|---|---|---|
| Real tools, LLM judge | 96.7% | 87.5% |
| Sim tools, LLM judge | 91.3% | 87.5% |

Table 2: Accuracy of task success/failure judgment on BFCL-Multi-Turn-Base. We compare the use of real tools and simulated tools. We use an LLM as judge. Both methods have good accuracy, while using simulated tools has 5.4% lower true positive rate.

provided in the Appendix A.3. Gecko has 100% accuracy in finding correct arguments and detecting syntactic errors for all the three helper LLMs. When there are semantic errors in the tool arguments, Gecko can detect 60% ∼ 70% of them. As shown in Fig. 4(a), accuracy drops by 1.5% if argument validation is removed. Results clearly demonstrate the usefulness of this component.

## 3.2 LLM-based tool response generator

Beyond using validated arguments and the tool schema (tool description, input schema, and output schema) as inputs, the response generator models the functionality of the tool, producing semantically realistic and schema-compliant outputs. To support multi-turn tool response synthesis, we additionally condition the generator on the current task state (Section 3.3), a compact state that grounds prior tool calls to prevent factual conflicts with earlier tool responses and to preserve cross-call consistency. Responses are produced by a helper LLM; prompts are provided in Appendix A.4. Examples are shown in Fig. 2 and Fig. 3(c).This is done through a helper LLM. Prompts are provided in the Appendix A. The responses are part of the input to task state estimator and are an important source of Gecko feedback. Examples of tool responses are shown in Fig. 2 and Fig. 3(c).

It is non-trivial to directly measure the effectiveness of response generation because there is no ground truths. To do so, we designed an indirect experiment where we obtain ground truths of task success/fail using real tools and rule-based task state matching on BFCL-Multi-Turn-Base. We compare the use of real tools and simulated tools in task success/failure estimation. Results are shown in Table 2. Compared with using real tools, tool simulation results in 5.4% accuracy decrease when deciding successful tasks as successful. Interestingly, regardless of using real or simulated tools, we observe the same accuracy of 87.5% in deciding task failure as failure. This experiment indicates that simulated tools and their responses are slightly more erroneous than real tools, but the overall accuracy of task success/failure estimation is acceptable.

### 3.3 LLM-BASED TASK STATE ESTIMATOR

Task state records the progress of a task. It summarizes the cumulative effects of past tool calls and serves as Gecko's central reference for grounding tool calls. Given the previous task state and the newest tool call and its response, a helper LLM updates the task state to reflect the effect of that call. For example, given the previous task state 'apples in cart: 3' and a tool response 'successfully delete one apple from cart', the updated state would be 'apples in cart: 2'. The most recent task state is used as input to the response generator (Section 3.2). For more examples, see Fig. 3(c) and Fig. 2.

Similar to task response generation, there is no direct measurement of task state estimation performance. In Table 2, the performance of task success/failure estimation is generally very good. This indirectly supports the effectiveness of task state estimation.

### 3.4 LLM-BASED TASK FEEDBACK GENERATOR

We use a judge LLM to create the feedback to be sent to the planning LLM. Two steps are involved. First, we use the task description as input and let the judge LLM generate a checklist specifying what aspects are important for indicating the completion of this task, such as 'The temp folder is created in document' and 'The report.pdf is moved to temp' for the task in Fig. 2. Second, we let the judge LLM decide whether the Gecko execution results fully satisfy the checklist: if yes, then the task feedback indicates success; if not fully satisfied, the judge LLM identifies remaining objectives, and another round of LLM planning and Gecko simulation will be executed. Specifically, the input to the judge LLM includes the task, tool calls from the planning LLM, the simulated responses (Section 3.2), and the task state (Section 3.3). An example of task feedback is shown in Fig. 3(c).

In Table 2, we present the accuracy of task success/failure judgement on the BFCL-Multi-Turn-Base. When using real tools, the LLM judge achieves a true positive rate of 96.7% and a true negative rate of 87.5%. This indicates that our task feedback generator works well.

The components described from Section 3.1 to Section 3.4 allow us to finally ground the agent tool calls. That is, after receiving the tool calls from the planning LLM, Gecko checks argument validity, simulates tool responses, estimate task state, and then give task feedback.

### 3.5 LLM-BASED API SCHEMA CONVERTER FOR VARIOUS TOOLS

Tools that have a standard OpenAPI schema can be directly used in Gecko. For Python functions and other non-OpenAPI tool definitions, Gecko uses an LLM-based schema converter to produce OpenAPI schemas. Given a description of a tool that details its purpose, input parameters, and expected output, an LLM generates an OpenAPI 3.1.0 specification in the JSON format. Our text prompt is provided in the Appendix A.7. This automated conversion allows Gecko to quickly integrate and simulate tools and thus support more tools than StableToolBench. The generated schemas are used by the argument validator (Section 3.1) and response generator (Section 3.2).

## 4 GROUNDING AGENT TEST-TIME SCALING (GATS)

Given a task and tool calls generated by the planning LLM, Gecko provides three types of feedback: argument validation, tool responses, and task feedback. In implementation, argument validation happens before response generation and returns validation feedback to the planning LLM immediately. If a tool call is valid, the simulated tool response is also returned immediately. From the planning LLM's perspective, Gecko mirrors real tools: each call either yields an error message from validation or a tool response. After each valid call, Gecko updates the task state aligned with the user-desired task. Task feedback is produced after the planning LLM finishes its output for the task. The LLM judge (Section 3.4) evaluates the latest task state and the sequence of tool calls against the task, determining success or identifying remaining objectives. This completes one attempt in GATS. If the attempt fails, the tool call sequence and the task feedback will be sent to the planning LLM in the next retry. Thanks to Gecko's task state recording and session-based isolation mechanism (Appendix A.2), retries can restart from exactly the same state snapshot, without interference from tool calls generated in previous attempts. A diagram of this iterative scaling method is drawn in Fig. 2. The pseudocode is provided in the appendix (Algorithm 1).

## 5 DISCUSSIONS

**Can Gecko be implemented only by prompting?** Technically yes, if we can find a *perfect* prompt to let the LLM output all the feedback and responses. However, the perfect prompt is almost impossible to create, because 1) our system is a combination of rule and LLM use, and 2) even if the prompt is successfully written, it will be too complex and hard for a LLM to understand.

**Why not use responses from real tools in Section 3.2?** While real tool responses are accurate, using them during test-time scaling has a few drawbacks. **First**, real tool execution may incur substantial cost, including computational overhead and API usage fees (e.g., RapidAPI charges per request). **Second**, performing iterative refinement directly on real tools increases the risk of unintended side effects, such as sending wrong emails.

**New research possibilities enabled by Gecko. First**, Gecko is complementary to existing tool-call data synthesis pipelines Liu et al. (2024b;a) as a verifier to improve dataset quality. A typical tool-call data point contains a task, tool definitions, and a tool-call sequence. These could be fed into Gecko, which would simulate tool responses, estimate task state, and return task feedback indicating whether the tool-call sequence solves the task. This feedback can be used to filter out or correct erroneous data points. **Second**, Gecko can turn existing SFT tool-call datasets into RL environments. Given tool definitions, Gecko can form an action space by converting each tool definition to a callable tool. After each action, Gecko returns an observation that simulates the tool execution result. Rewards are produced by an LLM judge via checklist–state comparison, allowing multiple valid action sequences and yielding fine-grained, stepwise signals for reward-function design. The resulting trajectories (states, actions, observations, rewards) can be used for offline RL, and the same interface supports online exploration in Gecko.

**Comparison with StableToolBench (STB) Guo et al. (2025).** While both Gecko and STB can simulate API responses, Gecko has a few key advantages. **First**, to simulate an API, STB needs to collect real responses from this API, which can be costly and less flexible. In comparison, Gecko directly supports new APIs using only API descriptions. **Second**, API-call data in STB has quality issues: it contains many erroneous responses due to invalid API calls, timeout errors, and server-side unavailability (30-40% error rates in our preliminary investigation). Gecko does not have these issues because the responses are simulated based on our converted OpenAPI schemas. **Third**, STB lacks API argument validations and generates responses for all API calls, including invalid ones, whereas real-world API servers reject such invalid calls. In contrast, Gecko has an argument validity checker that rejects invalid API calls and returns meaningful error messages, thus precisely simulating real-world API server behavior. **Finally**, STB focuses on individual API calls without considering multi-turn conversation history, while Gecko considers history from both task states and conversation, especially in multi-turn scenarios, which ensures logical coherence.

**Limitations.** Gecko currently only supports text-out tools, such as `get_temprature` and does not yet support tools that produce non-text outputs, whose outputs are non-text media, such as `download_video`. In addition, for tools that rely on complex, dynamic external databases, such as airline reservation systems, simulation outputs (*e.g.*, available flights or a user's booked tickets) may diverge from the real-world state. For real-world deployments of GATS, a possible mitigation is a hybrid execution mode: simulate state-changing (write) tools within Gecko while directly hosting read-only/query tools without simulation. This reduces simulation–reality drift for information retrieval while retaining Gecko's sandbox benefits for test-time scaling.

## 6 EXPERIMENTS

### 6.1 EXPERIMENTAL SETUP

**Benchmark.** We evaluate Gecko and the grounding agent test-time scaling (GATS) method on the Berkeley Function Call Leaderboard (BFCL) and the $\tau^2$-bench. **BFCL** evaluates LLM tool-use ability in three categories: non-live single-turn, live single-turn, and multi-turn. Single-turn means that a task must be completed in one user–assistant round; non-live indicates that tasks and tools are designed by experts, while live indicates that tasks and tools are sourced from real-world scenarios. Multi-turn requires the model to plan and generate tool calls across several rounds based on tool-execution results and the user feedback. Within single-turn, there are four task types: simple (one

Table 3: Method comparison on BFCLv3. We select eight most important metrics from the BFCL website. Overall accuracy is computed as the average of average 'Non-live single turn', average 'Live single turn', and 'Multi-turn' categories. GATS consistently improves various planning LLMs.

| Overall Acc | Model | Non-live single turn | | | | Live single turn | | | Multi-turn |
|---|---|---|---|---|---|---|---|---|---|
| | | simple | parallel | multiple | irrelevance | simple | multiple | irrelevance | base |
| **State-of-the-art reference models** | | | | | | | | | |
| 73.12 | ToolACE-2-8B | 88.00 | 92.50 | 92.50 | 95.41 | 70.93 | 79.01 | 84.80 | 49.00 |
| 79.27 | watt-tool-70B | 98.25 | 85.50 | 94.00 | 84.16 | 86.04 | 83.47 | 68.48 | 68.00 |
| 80.96 | xLAM-2-70b | 94.75 | 92.00 | 94.50 | 83.33 | 77.13 | 71.13 | 74.48 | 77.50 |
| **Baseline models and our proposed method** | | | | | | | | | |
| 66.20 | GPT-4.1-mini | 91.50 | 84.50 | 88.00 | 78.33 | 79.45 | 70.94 | 68.70 | 40.00 |
| 73.84 | +GATS | 96.25 | 88.00 | 95.50 | 84.58 | 84.49 | 74.54 | 80.83 | 50.50 |
| 58.85 | GPT-4.1-nano | 82.25 | 78.50 | 75.00 | 80.83 | 65.11 | 58.97 | 72.22 | 32.00 |
| 67.59 | +GATS | 93.25 | 88.50 | 95.00 | 81.25 | 77.13 | 69.80 | 80.38 | 37.50 |
| 76.93 | GPT-4o | 92.75 | 92.50 | 92.50 | 84.16 | 81.00 | 78.53 | 78.45 | 61.00 |
| 84.62 | +GATS | 96.50 | 95.00 | 95.50 | 95.83 | 84.10 | 81.01 | 93.42 | 72.00 |
| 73.78 | Qwen-3-14B | 95.50 | 92.50 | 95.00 | 84.58 | 86.04 | 80.81 | 77.44 | 48.00 |
| 78.60 | +GATS | 96.75 | 93.50 | 95.00 | 92.50 | 87.59 | 83.00 | 91.50 | 54.00 |
| 66.44 | Gemini-2.5-pro | 86.25 | 69.00 | 86.00 | 91.66 | 77.90 | 62.20 | 89.68 | 39.50 |
| 70.44 | +GATS | 92.25 | 75.00 | 89.00 | 92.50 | 80.62 | 67.99 | 91.83 | 44.00 |
| 70.40 | Deepseek-V3 | 97.00 | 92.00 | 94.00 | 80.41 | 86.04 | 79.48 | 72.56 | 41.00 |
| 72.90 | +GATS | 97.25 | 92.00 | 95.50 | 83.75 | 88.75 | 81.76 | 78.79 | 43.50 |
| 61.94 | GPT-5-thinking | 78.00 | 84.00 | 76.00 | 92.91 | 61.62 | 57.45 | 89.70 | 33.50 |
| 66.08 | +GATS | 85.00 | 90.50 | 83.00 | 93.75 | 67.44 | 63.24 | 90.38 | 36.50 |

tool call is executed to answer a user query), multiple (multiple tool calls are executed sequentially to answer a user query), parallel (multiple tool calls are executed in parallel to answer a user query), and irrelevance (none of the provided tools is appropriate, so the correct behavior is to avoid tool use). In total, BFCL contains 3,633 tasks involving 8,578 tools. $\tau^2$**-bench** is specifically designed to assess agent abilities in the real world in the retail scenario ($\tau^2$-retail) and airline scenario ($\tau^2$-airline). In $\tau^2$-bench, the agent must communicate with an LLM-simulated user, call domain APIs and follow domain policy rules (*e.g.*, refund and booking rules) to complete tasks. $\tau^2$-retail has 13 APIs and 114 tasks; $\tau^2$-airline has 12 APIs and 50 tasks.

**Evaluation metric.** For BFCL, we report *accuracy*, defined as the percentage of tasks completed correctly. For **single-turn** tasks, a prediction is counted as correct only if the tool calls produced by the model exactly match the reference solution. For **multi-turn** tasks, correctness is judged by comparing the task outcome after each turn, such as tools results and updated file contents, with the ground truth. A multi-turn task is considered successful only if the task outcomes match the ground truth at every turn. For $\tau^2$-bench, we use pass@1 averaged over 5 independent runs per task. Each task has an annotated goal database state, and a run is successful only if the agent responses provide all required information and the final database state matches the annotated goal.

**Implementation details.** We use GPT-4o-nano as helper LLM for argument validation because it is a relatively easy task. We use GPT-4o as helper/judge LLM response generation, task state estimation and task feedback generation, because these tasks are more complex. For BFCL, we execute GATS with a maximum of 3 times of retry in Gecko and directly use the resulting tool call sequence as the final answer for each task. On $\tau^2$-bench, because the internal database contents are not exposed to the agent and must be discovered via the native tools in $\tau^2$-bench (*e.g.*, prior reservations or available flights), simulated tool calls of Gecko may differ in details. To bridge this gap, for each user message, we first run GATS to generate up to 3 rollouts, then pass all attempts (including failures) and their task feedback as in-context examples to the $\tau^2$-bench agent, so the agent can learn to avoid potential errors in the $\tau^2$-bench environment. In our runs, the user simulator is configured to GPT-4.1. For result stability, we repeated the pass@1 evaluation five times and report the mean. The LLM temperature is fixed to 0 for all requests if applicable.

## 6.2 MAIN EVALUATION

**GATS consistently improves tool call capabilities of existing LLMs.** On BFCL and $\tau^2$-bench, we use various existing LLMs as the planning LLM, such as GPT-4.1-mini OpenAI (2025a), Deepseek V3 DeepSeek-AI et al. (2025), watt-tool-70B watt-ai (2025), Qwen3-14B Yang et al. (2025), Kimi-K2-Instruct Team et al. (2025), Claude Opus 4 Anthropic (2025) and GPT-5 OpenAI (2025b). We apply the proposed GATS on top of these planning LLMs and demonstrate the performance gain in Table 3 and Table 4. We have three observations.

**First**, GATS and consistently improves tool-call performance of these LLMs. For example, on BFCL, the overall accuracy of GPT-4o and Qwen-3-14B is improved from 76.93% and 73.78% to 84.62% and 78.60%, respectively. On $\tau^2$-bench, our method improves GPT-4o from 54.3% to 56.7%, and GPT-5-thinking from 71.0% to 72.9%. We also note that there is less improvement on $\tau^2$-bench. The reason is that $\tau^2$-bench tasks provide the agent with much less contexts, such as available flights, than BFCL, making it much more challenging for Gecko to simulate accurate responses (see Limitations in Section 5).**Second**, our method is effective for both single-turn and multi-turn tasks. For example, the improvement of GPT-4o on 'Live single turn' is +3.10% and +2.48% for 'simple' and 'multiple', respectively, while its improvement on 'Multi-turn' tasks is +8%. **Third** and interestingly, while some planning LLMs have different performance on single-turn tasks, GATS may bring them to similar levels. For example, on 'Multiple' under 'Non-live single turn', the performance of GPT-4.1-mini, GPT-4.1-nano, Deepseek-V3, and GPT-4o becomes ~95% from 88%, 75%, 94%, and 92.5%, respectively. It suggests there exists some upper limit of Gecko or the benchmark itself (*e.g.*, annotation errors). We leave its explanation to future work.

Table 4: Method comparison on $\tau^2$-bench. We report success rate (%) under $\tau$-retail and $\tau$-airline subsets and their average accuracy (Overall).

| Model | $\tau^2$-retail | $\tau^2$-airline | Overall |
|---|---|---|---|
| **State-of-the-art reference models** | | | |
| Claude Opus 4 | 81.8% | 60.0% | 70.9% |
| Claude Sonnet 4 | 75.0% | 55.5% | 65.25% |
| Kimi-K2-Instruct | 70.6% | 56.5% | 63.55% |
| **Baseline models and our proposed method** | | | |
| GPT-4o-mini | 46.1% | 28.4% | 37.3% |
| +GATS | 48.4% | 30.8% | 39.6% |
| GPT-4o | 63.7% | 44.8% | 54.3% |
| +GATS | 65.8% | 47.6% | 56.7% |
| GPT-5-thinking | 81.2% | 60.8% | 71.0% |
| +GATS | 82.6% | 63.2% | 72.9% |

**Comparison with the state of the art.** We apply GATS to GPT-4o and report new state of the art on BFCL: **overall accuracy = 84.62%**, which is +3.66% higher than xLAM-2-70B. Moreover, on various subsets, GPT-4o+GATS also reports very competitive performance, e.g., 96.50% on simple single turn, 95.00% on parallel single turn and 95.50% on multiple single turn. The multi-turn-base performance, 72.00%, is the second best among all the methods. For $\tau^2$-bench, GPT-5-thinking+GATS achieves **an overall accuracy of 72.9%**, which indicates state-of-the-art performance.

## 6.3 FURTHER ANALYSIS

**Ablation studies.** Gecko contains four key components: argument validation, task state estimator, response generator, and feedback generator. Among the four, response generation cannot be removed[3], so our ablation studies are for the rest three. We experiment on the BFCL-Multi-Turn-Base with GPT-4o. Results are shown in Fig. 4 (a). *w/o arg validation* removes both rule-based and LLM-based argument validation in Gecko. *w/o task state est.* does not estimate the task states. *w/o task feedback* replaces the judge LLM with a naive gating mechanism: if tool calls are generated, we give a success feedback; if no tool calls are generated, then failure feedback. Results show that removing argument validation slightly decreases accuracy (from 72.0% to 70.5%), while removing task state estimation has a greater impact (68.0%). Eliminating task feedback causes the most significant drop (61.5%), indicating that iterative feedback is most important when solving multi-turn tasks.

**Comparing different LLMs used in different components in Gecko.** To investigate the impact of using different helper/judge LLMs, we replace them with a different LLM while keeping the other components unchanged. Results on BFCL-Multi-Turn-Base are shown in Fig. 4(b), where the performance of the default setting is 72.0%. Replacing GPT-4.1-nano in argument validation with GPT-4o results in a very minor drop (71.5%). Because argument validation is relatively simple, it

---

[3]Nonetheless, Table 2 presented a variant of response generator by replacing simulated tools with real tools. We observe that simulated tools have a reasonably lower but acceptable true positive rate.

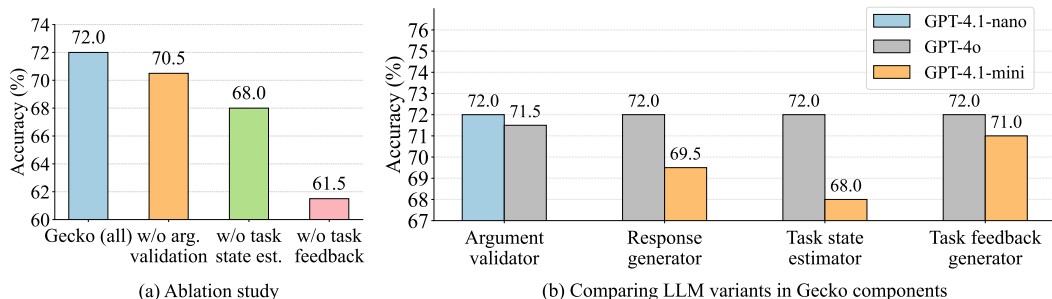

(a) Ablation study

(b) Comparing LLM variants in Gecko components

Figure 4: (a) Ablation study of Gecko on BFCL-Multi-Turn-Base. The full system (72.0%) is compared against variants with one component removed: argument validation (70.5%), task state estimation (68.0%), and task feedback (61.5%); (b) LLM replacement study on Gecko evaluated on BFCL 'Multi-turn base'. For each component, the bar on the left is the original performance 72.0%. Under LLM replacement, *e.g.*, replacing GPT-4.1-nano with GPT-4o for argument validation.

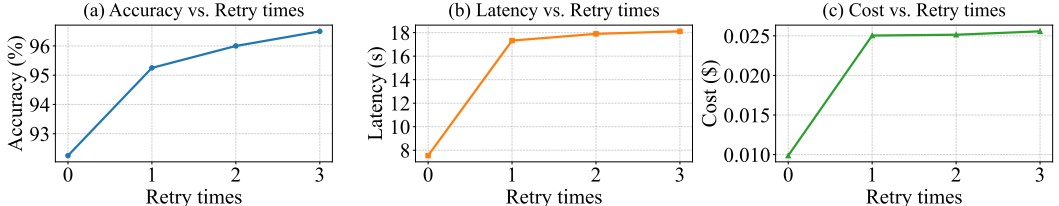

Figure 5: Test-time scaling behaviours. We evaluate GATS on the BFCL-Non-Live-Simple, using GPT-4o as the planning LLM. *Retry times* is the maximum number of feedback-based refinement steps allowed in GATS (Section 4). We report (a) accuracy (%), (b) average latency (s) and (c) average cost ($) per user task versus maximum retry times.

does not require strong LLMs like GPT-4o. If we replace GPT-4o with GPT-4.1-mini in response generation and task state estimation, performance drops from 72.0% to 69.5% and 68.0%, respectively. For the task feedback components, replacing GPT-4o with GPT-4.1-mini leads to a small decrease in performance (-1.0%). It shows the robustness of this component to weaker LLMs.

**Scaling behavior of GATS.** GATS allows a maximum of times of retry, where each retry includes Gecko feedback and then tool call refinement. We examine how the retry budget affects performance on the BFCL-Non-Live-Simple, where maximum retry times vary from 0 to 3. As shown in Fig. 5, increasing the max retry times improves accuracy, from 92.25% with no refinement to 96.50% with three times of refinement. Most accuracy gain comes from the first retry, while further retries add less improvement. Accordingly, latency and cost increase with retry times: runtime increases from 7.54 s to 18.11 s, and cost from $0.00987 to $0.02557. Both also demonstrate decreasing margin: most user tasks are resolved in the first retry, so they will not use up the maximum retry times. These results clearly demonstrate a trade-off between accuracy gain and cost.

# 7 CONCLUSION

This paper introduces Gecko, a comprehensive simulation environment that takes tool calls from planning LLMs as input and outputs a variety of feedback. Early feedback is the validity of tool calls, while task-level feedback considers simulated responses and an estimate of the task state. Building on Gecko, we propose a test-time method that iteratively refines tool calls via feedback, named GATS. Our method is shown to consistently improve the performance of various LLMs on agent tool call benchmarks. In the future, Gecko can serve as foundational infrastructure for agentic tool use, enabling the community to (1) improve agents' tool use at test time via tool simulation and stateful, task-aware feedback; (2) synthesize higher-quality tool-call data for supervised fine-tuning (SFT) by using Gecko as a verifier or within the data-synthesis loop; and (3) use Gecko to turn SFT tool-call datasets into reinforcement learning environments.

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

# A APPENDIX

## A.1 THE USE OF LARGE LANGUAGE MODELS (LLMS)

OpenAI ChatGPT was used for grammar checks and phrasing suggestions. Anthropic Claude (Claude Code) was used to assist with implementing and debugging portions of the code. LLMs did not contribute to research ideation or experimental design.

## A.2 DESIGN PRINCIPLES OF GECKO

Gecko is a simulated tool execution environment built on modern web-service principles and accessible over the network. It exposes a RESTful HTTP interface so clients can interact with it using standard JSON payloads.

Gecko uses **FastAPI**, a high-performance asynchronous web framework, to handle API requests, and **CAMEL** Li et al. (2023) as the LLM agent framework. FastAPI's native async support and lightweight routing let Gecko handle many concurrent requests with low latency.

The architecture follows a **middleware-based design pattern** that provides a clear separation of responsibilities. Each incoming request passes through a chain of middleware components that handle different aspects of the processing pipeline. The session middleware manages state isolation between different execution contexts, while the route middleware handles the mapping between API endpoints and their corresponding OpenAPI specifications. This layered approach ensures that each component focuses on a specific responsibility while maintaining loose coupling between different parts of the system.

To support concurrency and retry mechanisms, we designed a **session-based state management system**. Each client interaction runs inside its own session with a unique session ID. The session holds its own configuration, execution history, and task state history so the server can safely retry actions and replay earlier steps without mixing data from different clients. This isolation prevents users from interfering with each other and lets Gecko handle many users at once. Sessions persist across multiple tool calls, enabling multi-turn interactions that remain tied to the same session.

## A.3 ARGUMENT VALIDATOR IMPLEMENTATION DETAILS

### A.3.1 SYNTACTIC VALIDATION RULES

Our syntactic validation enforces comprehensive rule-based checks to ensure argument correctness at the structural level.

**Presence of required fields.** We verify that all fields marked `required` in the OpenAPI schema are present in the tool call. We also check that the tool call does not include any fields that are not defined in the schema. Tool calls that violate this rule will receive an instant error message.

**Type checking.** Each provided argument is checked against the type (e.g., `integer`, `float number`, `string`, `boolean`, `array`, and `object`) declared in the OpenAPI schema. Type mismatches will result in an error message.

**Constraint enforcement.** The validator enforces constraints such as numeric bounds (`minimum`/`maximum`, `exclusiveMinimum`/`exclusiveMaximum`), enumeration (`enum`), and length/pattern restrictions for strings (`miniLength`/`maxLength`). When constraints are violated, the validator reports the specific constraint failure.

### A.3.2 SEMANTIC VALIDATION VIA LLM

The semantic validator takes two primary inputs: the tool call to be verified and the corresponding OpenAPI schema. The validator returns a JSON-parsable message that indicates whether the tool call is semantically acceptable and enumerates any problems found. The main prompt for the semantic validator is provided below.

```
Please validate the given function call arguments against their
    parameter schemas.

**Validation Rules**:
1. **Scope**
   - Only validate arguments defined in the provided schemas.
   - Ignore arguments not present in the schema (do not treat them as
   errors).
   - Type validation has already been handled elsewhere. Just skip
   type checking.

2. **Semantic Checks**
   - Validate according to the parameter description, examples, enums,
    or format requirements.
   - If examples are provided (e.g. "full-time, part-time"), treat
   them as semantic categories. Any value in the same category (e.g.
   "internship", "contract") is valid.
   - If the description specifies a format (e.g. `YYYY-MM-DD`),
    enforce that exact pattern.
   - Use common sense to ensure values are within a reasonable range (
   e.g. interest rate within [0,1]; clock hour within [0,12]).
   - Detect redundant/overlapping information across arguments (e.g. `
   item="large pizza"` and `size="large"` are overlap).
   - If uncertain about validity, default to considering the argument
   valid.

3. **Error Messages**
   - Concise, precise, and human-readable.
   - Do not include or suggest correct values.
   - Only state which argument is invalid and why.

**Output Format**:
```

valid=\<true|false> error\_message="\<if false, list each invalid
    argument and reason>"

```

**Example**:
- **params_schema**
  `[{"location": "The city that you want to go, e.g. 'Beijing, China
  '"}, {"date": "The start date for the booking, format: YYYY-MM-DD
  "}]`
- **args**
  `{"location": "London", "date": "01/01/2024"}`
- **Output**
  `valid=false error_message="location not in required format (should
   include city and country); date not in required format (YYYY-MM-DD
   )"`
```

## A.4 RESPONSE GENERATOR IMPLEMENTATION DETAILS

The response generator synthesizes a tool response given three inputs: a tool call, the corresponding OpenAPI schema, and the current estimated task state. The main prompt for the response generator is provided below.

```
You are an API simulation engine that generates JSON responses
    strictly following OpenAPI 3.1 schemas.

Rules:
1. Schema first. Always match the schema exactly (structure, names,
    types, formats, required fields).
2. Entity-level consistency. Do not contradict any provided state or
    any prior successful responses in this session.
3. Open-world reads. For read/query/search operations, if requested
    entities/data are absent in the provided state, you MUST
    synthesize realistic, schema-compliant values instead of returning
     not-found or error responses.
4. Writes remain consistent. For create/modify/delete operations,
    produce a success result consistent with the schema unless it
    would contradict previously returned state; do not invent
    conflicts.
5. No extra rules. Do not invent constraints beyond the tool
    definition and the provided state.

Realism & uniqueness guidelines (domain-agnostic):
- Deterministic diversity: derive identifier-like fields using stable
    transforms of input arguments (e.g., incorporating parts of
    arguments or their hashes) so that different arguments yield
    different values within the session, while the same arguments
    yield stable values.
- Identifier-like fields (e.g., keys ending with `_id`, `Id`, `code`,
    `number`): prefer distinct values for distinct entities in the
    same response unless the schema indicates they refer to the same
    entity.
- Consistency: when two items share the same identifier-like value in
    one response, their associated attributes MUST NOT contradict each
     other within that response.
- Plausible formats: choose values that look realistic when the schema
     allows free-form text, but always prioritize matching schema
    types and formats.
- Temporal consistency when both present: end/arrival timestamps
    should be after start/departure timestamps; choose plausible
    intervals without assuming domain-specific constraints.
- Diversity: vary counts and enumerations when optional, within
    reasonable ranges, while staying schema-compliant.

Illustrative synthesis examples:
- Tool name: get_user_details:
  Request: {"user_id": "john_doe_001"}
  Response (success object):
  {
    "user_id": "john_doe_001",
    "name": "John Doe",
    "email": "john.doe@example.com",
    "phone": "+1-222-345-6789",
    "loyalty_status": "silver",
    "miles": 50000,
    "address": "7340 Oak Street, San Francisco, CA 94110"
  }
```

## A.5 TASK STATE ESTIMATION IMPLEMENTATION DETAILS

The task state estimation contains two phases: initialization and progressive updating. First, the task state bootstrapper constructs an initial task state from the task description and the relevant OpenAPI schemas. The task state updater then progressively revises this state as the response generator synthesizes tool responses.

The main prompt for the task state bootstrapper is as below.

```
You are initializing the system state for a task execution system with
    multiple toolkits based on the given background information.
IMPORTANT: System state should contain ONLY these two types of data:
1. **Domain Data (Databases)**: The actual data that tools operate on
    * FileSystem toolkit: files, directories structure
    * Airline toolkit: users, flights, tickets, bookings
    * Message toolkit: messages, inbox items
    * These are stored at appropriate top-level or domain-specific keys
2. **Runtime Variables**: Execution context and session state
    * Store these DIRECTLY under 'runtime\_state' (flat structure)
    * Examples: current\_working\_directory, current\_user, is\_logged\
    _in, session\_token
    * IMPORTANT: Read toolkit descriptions carefully for initialization
      requirements

CRITICAL:
* NO 'runtime\_state.toolkits' structure - keep runtime\_state FLAT
* NO nested toolkit sections within runtime\_state
* NO duplicate concepts (e.g., only ONE current\_directory for the
    whole system)
* NO static values, validation rules, or schema metadata

Example of CORRECT runtime\_state structure:
"runtime\_state": {{
"current\_working\_directory": "/root"
}}

Rules:
1. Preserve all existing structures in the backgound information
2. Add runtime variables DIRECTLY under 'runtime\_state' (flat
    structure)
3. Add domain data at appropriate keys (not in runtime\_state)
4. NEVER create 'runtime\_state.toolkits' or any similar nesting
5. Avoid duplicating the same concept
6. Output valid JSON only

Background information:
{background_information}

Toolkits summary:
{json.dumps(toolkits\_summary, indent=2)}

Return the UPDATED config JSON with necessary domain data and runtime
    state.
```

The main prompt for the task state updater is as below.

```
You are an expert at tracking the execution state of a task.
Update the system state based on the tool calls and their effects on
    the system.

IMPORTANT GUIDELINES
1. **State Tracking Principles**
   - Update the system state to reflect ALL persistent state changes
   caused by tool calls
   - Operations that create, modify, or delete resources MUST update
   the corresponding structures
   - {"In synthesis mode Store ALL synthesized data from read
   operations as ground truth state" if synthesis_mode else "
   Operations that just query or read data should NOT add their
   results to the system state"}
2. **system state Organization**
   - When tool operations modify existing structures, update them
   directly (e.g., adding a new directory should add it to the
   directory tree)
   - For execution context that doesn't fit existing domain structures
   , use the root-level "runtime_state"
   - The "runtime_state" section is ONLY for execution context and
   ephemeral telemetry (e.g., current location/cursor, active
   selections, session info, temporary counters)
   - DO NOT store canonical domain data in "runtime_state" (e.g.,
   files, inbox messages, database rows must live under their domain
   keys)
   - If a counter already lives under "runtime_state" (e.g.,
   runtime_state.toolkits.messageapi.message_count), update it ONLY
   when the tool call semantics deterministically imply the change;
   never infer from read-only calls
   - Never duplicate the same fact both in a domain section and in "
   runtime_state"
3. **Value Formatting**
   - When recording locations, positions, or identifiers, use complete
   , unambiguous values
   - Avoid partial or relative references that could be misinterpreted
   - Preserve the format conventions used in the original system state
4. **What Changes to Track**
   - Resource creation/deletion/modification (files, directories,
   database records, etc.)
   - State transitions (status changes, position changes, mode
   switches)
   - Context updates (current location, active selections, session
   data)
   - DO NOT track query results, search results, temporary
   computations, or read-only operation outputs
5. **Example Structure with runtime_state**
   {{
     "DomainSystem" {{
       // Domain resources with any modifications from state-changing
   operations
     }},
     "runtime_state" {{
       // Execution context only (no canonical domain data)
       "current_context" "...",
       "current_user" "USR001"
     }}
   }}

Output the updated system state in JSON format only.
```

### A.6 TASK FEEDBACK GENERATION IMPLEMENTATION DETAILS

Task feedback generation has two steps. First, given the task description, we generate a detailed checklist that decomposes and clarifies the task intent into clear and verifiable objectives. Second, the checklist is verified item by item by an LLM-based judge, which receives the checklist, current task state, executed tool calls, the response from the planning LLM, and the corresponding OpenAPI schemas. The judge aggregates any checklist objectives that fail verification into a compact task-level feedback message, which is returned to the planning agent to guide subsequent refinements.

The main prompt for checklist generation is given below.

```
You produce ATOMIC, STATE-OPERATION-BASED verification checklists for
    tasks in ANY domain.
PREVIOUS TASKS (assumed done; resolve references only, do NOT re-
    verify): {prev_text}
CURRENT TASK: {current_task}

CRITICAL MULTI-TURN CONTEXT RULE:
When task mentions "values obtained", "results from previous", or
    specific counts like "three values", these refer to OUTPUT from
    the LAST task in PREVIOUS TASKS, not data from earlier tasks

RULES
1) Verify ONLY the current task. Return the MINIMAL set; if one item
    suffices, return EXACTLY one.
2) Each item = one pass/fail assertion about final state (implied by
    the question, do not guess the answer by yourself) or an executed
    operation (no procedures).
3) IDENTIFY ALL SEMANTIC UNITS: Each complete thought, question (
    direct or indirect like "I wonder"), or action in the task needs
    verification
4) PRESERVE LOGICAL FLOW: When actions depend on prior information or
    results, verify each step
5) Use explicit identifiers/paths/IDs when inferable; avoid vague
    pronouns.
6) RESOLVE AMBIGUOUS REFERENCES: When the current task contains
    pronouns like "the file", "it", "that item", etc., resolve them to
     specific entities based on PREVIOUS TASKS context.
7) Do NOT add optional behaviors (saving/exporting/logging/formatting)
     unless explicitly required.
8) Search/lookup/filter. Assert the search was executed with the
    specified term/criteria; do NOT require matches unless asked.
9) Transform/update. Assert the stated post-condition holds; do NOT
    invent extra artifacts.
10) Copy operations. Verify: source file remains intact (copy
    preserves original); destination file exists with the new name.
11) Create/Delete to assert existence/absence as specified.
12) Discrete relocation between containers (domain-agnostic). If
    applicable and implied: destination container exists (if mentioned
    ); entity absent at source (for move, not copy); entity present at
     destination.

OUTPUT
Return ONLY a pure JSON array of objects with a single key "
    description". No extra text.
```

The main prompt for the LLM judge is as follows.

```
You are an expert in verifying a checklist based on the execution
    results.
You have access to:
1. Current system state: The system state after execution
2. Tool calls: The list of functions that were called WITH their
    results
3. Agent response: The agent's output/explanation (if available)
4. Conversation history: Previous turns showing the context of how
    data was obtained{history_text}

IMPORTANT: Some operations (like grep, sort, find, ls) are query
    operations that don't modify the system state.
For these operations, verify their execution by checking if the
    corresponding tool was called in the tool_calls list OR Looking
    for evidence in the agent_response (if provided) that the
    operation was performed and results were obtained

Guidelines:
1. Verify items in the checklist one by one
2. For state-modifying operations (mkdir, create file, cd), check the
    system state for changes
3. For query operations (grep, sort, ls, find), check tool_calls and
    agent_response
4. For efficiency-related checklist items, analyze whether multiple
    tool calls could be merged based on the tool definitions provided
5. Status should be one of ["success", "failed", "unknown"]
"success": Task completed (evidence in tool_calls/system state/
    agent_response)
"failed": Task NOT completed AND agent provided NO explanation
"unknown": Task NOT completed BUT agent explained why (e.g., "missing
    information", "need user confirmation", "tool unavailable")
6. Just modify the status and reasoning fields of the checklist items,
     do not include any other text outside the json.{tool_defs_text}
7. If a tool call is made as the task required, do not mark it as
    failed even if the result is not as you expected
8. Use conversation history to understand data references.
9. When evaluating relevance, consider the full multi-turn context to
    understand where numbers/data come from

Evaluation principles (keep these high-level and tool-agnostic):
Only mark "failed" when there is clear evidence that the requirement
    was not met and no explanation was provided by the agent; if
    evidence is incomplete or you are not sure, use "unknown".
Accept equivalent pipelines that produce the required final outcome,
    regardless of operation order or scope.
Do not fail solely because an intermediate step operated on a broader
    scope; fail only if the final required subset/result is missing or
     incorrect.

The output should be in json format:
[
    {{"name": "...", "description": "...", "reasoning": "...", "status
    ": "success" or "failed" or "unknown"}}
    {{"name": "...", "description": "...", "reasoning": "...", "status
    ": "success" or "failed" or "unknown"}}
    ...
]
Do not include any other text outside the json.
```

### A.7 LLM-BASED API SCHEMA CONVERTER IMPLEMENTATION DETAILS

The main prompt for the API schema converter is given below.

```
Convert this tool description to an OpenAPI 3.1 endpoint specification

Tool description:
{tool_description}

Create an endpoint object with these exact fields:
1. operationId: {tool['name']}
2. summary: one-line description (keep it short)
3. description: brief description of the operation (1 sentence max)
4. requestBody: proper schema based on parameters
5. responses: ONLY USE STATUS 200 with oneOf schema for success/error
   - Success response: Based on tool's PURPOSE (not generic "result")
   - Error response: Standard error object
6. Analyze the tool's PURPOSE to generate an appropriate response
    schema. For example:
If it calculates something (area, factorial, etc.), return the
    calculated value
If it fetches data (user info, list of items), return the data
    structure
If it performs an action (create, update, delete), return success
    confirmation with relevant details
If it searches/filters, return matching results

CRITICAL REQUIREMENTS:
- NEVER use $ref - always inline all schemas
- ONLY use HTTP status 200 for ALL responses
- Use oneOf schema in the 200 response to handle both success and
    error cases
- All properties MUST have a "description" field
- The requestBody must have a schema with type "object"
- If no parameters, still include requestBody with empty properties

Return ONLY the endpoint object JSON.
```

### A.8 EXPERIMENTS ON THE ACCURACY OF ARGUMENT VALIDATION

We evaluated the argument validator on the BFCL-Live-Simple subset. GPT-4o was used to generate tool calls for each example. For tool calls that passed the official BFCL evaluation, we marked them as correct. For tool calls that failed the BFCL evaluation, human annotators labeled each failure as either a *syntactic error* (violates the tool schema, e.g., missing required field or wrong field name/type) or a *semantic error* (schema-conformant but semantically inappropriate, e.g., wrong granularity or implausible value). These labels formed the ground truth.

We then ran our argument validator on the same generated tool calls and compared its predictions (correct / syntactic error/ semantic error) to the ground truth. Table 1 reports the detection rates.

### A.9 PSEUDOCODE OF GATS

**Algorithm 1:** GATS loop for an individual dialog turn (inline detailed comments).

**Input:** Task $T$, tool definitions $D$, schema converter $C$, planner LLM $\pi$, argument validator $V$, response generator $G$, task state estimator $E$, judge LLM $J$, initial task state $st_{\text{init}}$, maximum retries $R_{\max}$

**Output:** Tool-call sequence $trace$

$S \leftarrow C.\text{convert}(D)$ // convert tool definitions $D$ into schemas $S$

$checklist \leftarrow J.\text{gen\_checklist}(T)$ // decompose task $T$ to checklist

$last\_error \leftarrow \text{null}$ // holds last validation or judge error

**for** $attempt \leftarrow 0$ **to** $R_{\max}$ **do**
 $trace \leftarrow []$ // *trace* records tool call sequence
 $st \leftarrow st_{\text{init}}$ // initialize task state from previous task state
 **while** *true* **do**
  $c \leftarrow \pi.\text{gen\_next}(T, st, trace, last\_error)$ // planner generates tool call
  **if** $c == STOP$ **then**
   **break** // planner finished tool use for this attempt
  $(valid, errors) \leftarrow V.\text{validate}(c, S)$ // arguments validation
  **if** *not valid* **then**
   $last\_error \leftarrow errors$ // return validation feedback
   **continue**
  $r_c \leftarrow G.\text{generate}(c, S, st)$ // simulate tool execution
  $st \leftarrow E.\text{update}(st, c, r_c)$ // update task state
  append $c$ to $trace$ // record executed call
 $feedback \leftarrow J.\text{judge}(checklist, st, trace, S)$ // give task-level feedback
 **if** $feedback.success == true$ **then**
  **break** // task satisfied
 $last\_error \leftarrow feedback.\text{error}$ // return task-level feedback

**return** $trace$

