# OpenReview forum: "Gecko: A Simulation Environment to Ground Agent Tool Calls with Stateful Feedback for Refinement"
_ICLR.cc/2026/Conference — ICLR 2026 Conference Withdrawn Submission_

### Official Review · Reviewer_SkVK · 2025-10-17

**Soundness:** 2
**Presentation:** 3
**Contribution:** 2
**Rating:** 4
**Confidence:** 4

**Summary:**

The paper presents Gecko, an environment that simulates tool responses using LLMs. Gecko has multiple components to check argument validity, generate tool response, know the task state, create the feedback, and API schema converter. Building upon these, they introduce GATS, a test-time refinement that gets the feedback from Gecko to make better tool selections. They evaluated the entire pipeline on BFCL and τ2-bench and showed consistent improvements.

**Strengths:**

The research problem is timely given the rise of agentic LLMs.

The improvements over the base models are consistent and clearly demonstrated.

The paper is well written and easy to follow.

**Weaknesses:**

1. The Gecko pipeline consists of five LLM-based modules. While the overall improvements are promising, there is limited component-level evaluation. It remains unclear how much of the gain stems from the Gecko/GATS design versus potential artifacts of stacking multiple LLMs. Also, all components are implemented using GPT-family models, which raises concerns about potential GPT-specific bias.

2. The evaluation lacks comparisons to simpler baselines that attempt to achieve refinement through prompting alone. While the authors suggest that Gecko's functionality is hard to capture in a single prompt, this remains untested. Without such a baseline, it is difficult to assess how much of the observed improvements are due to the modular Gecko framework versus what could be achieved with a crafted prompt using a single LLM.

3. The paper lacks qualitative examples from benchmarks that illustrate how tool use improves over iterations. Without such examples, it is difficult to interpret the refinement process or clearly see where Gecko contributes to better tool selection and reasoning.

4. The set of open-source models used in the main experiments is somewhat limited. The paper lacks evaluations on models with more diverse architectures, sizes, and training objectives (like reasoning ones) which makes hard to assess how broadly Gecko’s improvements generalize.

**Questions:**

Check weaknesses for my main concerns.

In Table 3, I assume the reference models are results taken from the BFCL leaderboard and not run by the authors, right? If that’s the case, why do I see some inconsistencies when I check the leaderboard?

---

### Official Review · Reviewer_Vx59 · 2025-10-30

**Soundness:** 3
**Presentation:** 3
**Contribution:** 2
**Rating:** 4
**Confidence:** 4

**Summary:**

This paper presents Gecko, a simulation environment designed to help LLM agents refine their tool-use capabilities. Gecko validates tool-call arguments, generates realistic tool responses, and estimates task state, providing comprehensive feedback to a planning LLM. This feedback enables an iterative refinement method called Grounding Agent Test-time Scaling (GATS). The core contribution is a system that avoids the costs and risks of using real tools during test-time refinement. Experiments on BFCL and τ²-bench show that GATS consistently improves the tool-calling accuracy of various LLMs.

**Strengths:**

The system is well-designed and tackles a clear, practical problem: the financial cost and unintended consequences of testing tool-calling agents on real APIs. (To make this motivation even stronger upfront, the authors could give concrete examples of expensive or risky tools (e.g., a "send_email" or "place_order" API). This would immediately help the reader grasp why a simulated environment is necessary, even though it introduces its own computational costs. )

Also, I see some nice discussion points in the potential impact of Gecko,  especially for synthetic data generator for RL.

**Weaknesses:**

1. The validation of the LLM-based tool response generator relies heavily on the BFCL benchmark. It is unclear how well this approach generalizes to a wider array of tools with more complex and specific  outputs. The paper would be strengthened by a more explicit discussion on the boundaries of what kinds of tools Gecko can effectively simulate (e.g., read-only vs. state-changing) and which categories remain challenging.

2. I think the performance improvements from GATS are marginal (e.g., ~2% on τ²-bench). Given that GATS incurs substantially higher computational costs due to multiple LLM calls for validation, response generation, and feedback, a comparison against other iterative, cost-equivalent methods (e.g., self-reflection or chain-of-thought) is missing. This makes it difficult to fully assess whether the gains are due to the specific feedback from Gecko or simply the effect of additional compute and iteration.

***Minor Issues***

There are inconsistent citation formats in the text (e.g., "GPT-4o OpenAI et al. (2024), Qwen3 Yang et al. (2025)" should be "GPT-4o (OpenAI et al., 2024), Qwen3 (Yang et al., 2025)").

The "Discussions" section (Section 5) would be better positioned after the "Experiments" section (Section 6) to maintain a more conventional paper flow.

**Questions:**

1. Could you elaborate on the limitations of the response generator? Specifically, what categories of tools (e.g., those reliant on complex, live databases) are less suitable for simulation in Gecko

2. Given the non-trivial computational cost of GATS, how does its performance compare to a baseline that also uses alternative refinement strategies, like self-reflection?

---

### Official Review · Reviewer_4UB6 · 2025-10-30

**Soundness:** 3
**Presentation:** 3
**Contribution:** 2
**Rating:** 4
**Confidence:** 4

**Summary:**

This paper aims to address two core problems in tool-use for Large Language Model (LLM) agents: incorrect tool selection and invalid arguments, and the high cost and potential risks (e.g., unintended side effects) of using real-world tools for iterative refinement. To solve these issues, the authors propose a novel simulation environment Gecko, of which the core function is to receive tool calls planned by an LLM agent and provide three types of feedback. Based on the feedback loop provided by Gecko, the authors propose a test-time scaling method called GATS which allows the planning LLM to "trial-and-error" in the simulated environment, iteratively refining its tool-call sequence based on Gecko's feedback. Their experimental results show that GATS consistently improves the tool-calling performance of various LLMs, including GPT-4o, on the BFCL and $\tau^{2}$-bench benchmarks.

**Strengths:**

1. The paper clearly articulates a real problem: using real tools for iterative refinement is expensive and risky. The motivation for a safe, cost-effective sandbox is compelling and addresses genuine deployment concerns.

2. Gecko's three-layer design is thorough and well-validated.

3. The authors conduct extensive experiments across multiple benchmarks and models, verifying Gecko's advantage.

**Weaknesses:**

1. Simulated tools show 5.4% lower true positive rate than real tools, but the paper lacks the analysis of error accumulation in multi-turn scenarios and failure mode categorization.

2. While claiming cost savings, the authors do not account for the multiple GPT-4o calls within Gecko (response generation, state estimation, task feedback) and do not provide quantitative comparisons with real tool costs. Gecko operates by cascading calls to multiple expensive LLMs (like GPT-4o), of which the inference cost is very likely to be far higher than the cost of many real APIs.

3. As noted in 'Contribution', this work is more like a complex engineering pipeline rather than a novel methodological contribution.

**Questions:**

1. If Gecko requires a powerful LLM (like GPT-4o) as a "judge" to determine if a tool call is correct, why not just use powerful GPT-4o as the "planner" to generate the tool call in the first place? The paper shows that GATS still improves GPT-4o as a planner (from 76.93% -> 84.62%). But this raises a deeper question: Does this just prove that "iterative refinement with more tokens" is better than "single-pass generation"? This makes GATS look like an extremely expensive form of "self-consistency," with a token cost that may grow exponentially (each refinement requires the full Gecko LLM chain).

2. Figure 5c shows the cost of GATS, but it is not compared to the cost of "real tools". Can the authors provide a direct, end-to-end cost and latency comparison between GATS (including all internal LLM calls) and an identical refinement loop using real tools (e.g., those from the BFCL benchmark)?

3. The 5.4% judgment accuracy gap in Table 2 is concerning. How does this simulation error accumulate over multiple iterations of GATS? Is there a risk that an agent, refined on faulty simulation feedback, will actually perform worse in the real world?

4. Why τ²-bench shows much smaller improvements (+2.4% vs. +7.7% on BFCL)? Is this a fundamental limitation of Gecko's design or can it be addressed?

---

### Official Review · Reviewer_jNJW · 2025-11-01

**Soundness:** 4
**Presentation:** 4
**Contribution:** 3
**Rating:** 4
**Confidence:** 5

**Summary:**

Gecko is an environment designed to simulate tool executions and provide three types of feedback. First, it validates inputs against predefined formats, using a combination of rules and a helper LLM. Second, it simulates tool responses by carefully prompting a helper LLM with the validated tool call, the tool schema, and the current task state to ensure that the responses are semantically appropriate and consistent with prior tool calls. Third, Gecko estimates the key states that reflect task progress based on simulated tool responses. A judge LLM then assesses the inferred task state, checks whether the task objectives have been met, and provides feedback on completion status and any remaining issues. Additionally, these three types of feedback facilitate a tool-call refinement method called GATS (Grounding Agent Test-time Scaling). Through this method, feedback from Gecko is used to refine tool calls in the planning LLM, which are then sent back to Gecko for further feedback.

**Strengths:**

Simplicity: The approach is relatively simple, making it easy to understand and implement.

Improved Generalization: GATS appears to help bring the performance of different planning LLMs to a similar level on single-turn tasks.

Real Tool Responses vs. Simulated Responses: As the authors note, using real tool responses for test-time scaling has certain drawbacks, such as the potential for high computational cost, API usage fees (e.g., RapidAPI), and the risk of unintended side effects (e.g., sending incorrect emails). Gecko avoids these issues by simulating responses.

**Weaknesses:**

Comparative Analysis: A comparison study with structured decoding, particularly with an ablation study (e.g., Figure 4), would be valuable for grounding the net advantages of the approach and shortcomings if any.

Unclear Implementation Details: As noted in the questions section, it remains unclear how exactly Section A.3.1 is implemented and how effective the validation rules enforcement is in practice. More details on the efficacy of the implementation would strengthen the paper.

**Questions:**

In my opinion, one of the key contributions of this paper is the ability to enforce validation rules. However, upon carefully reviewing Section A.3.1, I still find it unclear how the mapping from prompts to the set of rules that need to be enforced is curated, who is responsible for curating it, and what the error bounds on these rules are. Basic validation tasks like checking for the presence of required fields and type validation are relatively straightforward, but I would have liked a more detailed explanation of how the rule enforcement is implemented. Moreover, constraint enforcement seems more complex, especially since a function can have any *arbitrary* kind of constraint. Are these constraints defined in the function schema? In the function documentation? Do they require decorator functions? Are all possible constraints parsed and enforced? I believe more clarity on these points would enhance the understanding of the approach's robustness.

---

### Note · Authors · 2026-01-07

I have read and agree with the venue's withdrawal policy on behalf of myself and my co-authors.